# Neuromechanical changes in Achilles tendinopathy and the effects of exercise-induced mechanical tendon loading: a protocol for a systematic review

Ignacio Contreras-Hernandez,[1] Deborah Falla ![ORCID],[1] Alessandro Schneebeli ![ORCID],[1,2] Eduardo Martinez-Valdes ![ORCID] [1]

[1]Centre of Precision Rehabilitation for Spinal Pain (CPR Spine), School of Sport, Exercise and Rehabilitation Sciences, University of Birmingham, Birmingham, UK
[2]Rehabilitation Research Laboratory, Department of Business Economics, Health and Social Care, University of Applied Sciences and Arts of Southern Switzerland, Manno, Switzerland

**Correspondence to**
Dr Eduardo Martinez-Valdes;
E.A.MartinezValdes@bham.ac.uk

## ABSTRACT

**Introduction** Achilles tendinopathy (AT) is a debilitating overuse injury characterised by pain, impaired functional performance, morpho-mechanical changes to the Achilles tendon and triceps surae neuromuscular alterations. Loading-based exercise has become the principal non-surgical choice for the treatment of AT; however, mechanistic evidence by which loading-based treatment may help to resolve tendon pain remains unclear. This systematic review aims to summarise the evidence of the neuromechanical changes produced by AT and by exercise-induced mechanical loading.

**Methods and analysis** This systematic review protocol was informed and reported in accordance with the Preferred Reporting Items for Systematic Review and Meta-Analysis (PRISMA-P) and the Cochrane Handbook for Systematic Reviews of Interventions. Pubmed, MEDLINE, EMBASE, CINAHL Plus, Web of Science and SPORTDiscus electronic databases will be searched from inception to February 2021. Additionally, grey literature and key journals will be reviewed. Risk of bias will be determined independently by two reviewers using the version 2 of the Cochrane risk-of-bias tool for randomised trials (RoB 2) and the risk of bias in non-randomised studies - of interventions (ROBINS-I) tool according to Cochrane recommendations. Quality of the cumulative evidence will be assessed with the Grading of Recommendations, Assessment, Development, and Evaluation (GRADE) guidelines. If homogeneity exists between groups of studies, a random-effects meta-analysis will be conducted. If not, results will be synthesised narratively.

**Ethics and dissemination** Our findings will be disseminated through publication in a peer-reviewed journal and presented at conferences. No ethical approval was required.

**PROSPERO registration number** CRD42021231933.

## INTRODUCTION

Tendinopathy describes a spectrum of pathological changes to the tendon, leading to pain and reduced function. The essence of tendinopathy is a failed healing response, characterised by abnormalities in the microstructure, composition and cellularity of the tendon,[1] with degeneration and disorganised

### Strengths and limitations of this study

► This will be the first systematic review to synthesise evidence of neuromechanical changes produced by Achilles tendinopathy (AT).
► This will also be the first systematic review examining the effect of exercise-induced mechanical loading protocols on neuromechanical changes observed in individuals with AT.
► In accordance with the Grading of Recommendations, Assessment, Development, and Evaluation framework, the inclusion of observational studies might reduce the quality of evidence.

proliferation of tenocytes, disruption of collagen fibres and subsequent increase in non-collagenous matrix.[2]

Achilles tendinopathy (AT) is a debilitating overuse injury which causes considerable morbidity and functional impairment among the athletic and general population.[3 4] Achilles tendon injuries can be separated into non-insertional tendinopathy (55%–65% of the injuries), insertional tendinopathy (20%–25%) and proximal musculotendinous junction injuries (9%–25%), according to the location of pain.[5] Nevertheless, individuals may present symptoms at the insertion and mid-portion concurrently, and approximately 30% have bilateral pain.[6] Morphological comparisons of tendinopathic and healthy tendons have demonstrated a larger cross-sectional area (CSA) for the degenerated Achilles tendon.[7] It is believed that this increase is due to an accumulation of water and increased ground substance as a result of the pathology.[8] In addition, changes to the tendon's stiffness has been reported.[7] Typically, a thicker tendon is considered mechanically stronger due to its ability to dissipate high stresses (force/area) across the tendon and yield

lower strain energy.[7] However, degenerated Achilles tendon has lower stiffness and Young's modulus compared with healthy tendons.[7]

Human movement emerges from the interplay among descending output from the central nervous system (CNS), sensory input from the body and environment, muscle dynamics and the whole body dynamics.[9] Thus, neuromechanics is the study of the coupling between neural information processing and mechanical behaviour.[9] Fundamentally, the CNS plans, initiates and sends motor commands to muscle,[10] then the muscle executes the motor command producing force to pull the tendon, and finally, the tendon transmits and modulates muscle force controlling the movement.[11] Additionally, peripheral components of this hierarchical system (ie, muscle spindles and Golgi tendon organs) send feedback signals to assist the CNS in motor commands planning.[12]

Tendons require the ability to withstand, store and then deliver substantial force to perform day-to-day activities.[13] During sports-related activities, where the repetition and speed of the loading are drastically increased, the mechanical force placed on the tendon becomes substantially amplified.[14] Therefore, a decrease in tendon stiffness will cause the muscle fascicles to shorten more to account for the increased compliance of the tendon.[7] This may limit the muscle's ability to function within the force–length curve's optimal region, thereby affecting movement economy.[7]

In addition to the morpho-mechanical changes induced by AT, several neuromechanical adaptations have been reported in individuals with AT.[11] Neuromechanical adaptations include how the mechanical system may offload a task that the neural system needs to accomplish[9] and it is possible only through a tight connection between sensory and motor systems. One of these neuromechanical adaptations is the electromechanical delay (EMD), which is defined as the time lag between muscle activation and the mechanical force produced, which dictates the muscle-tendon unit's temporal efficiency.[11] Longer EMD has recently been reported for the affected side compared with the non-affected side in individuals with Achilles tendinosis, indicating a compromised triceps surae musculotendinous unit temporal efficiency (ability to transmit force from the muscle to the tendon as quickly as possible).[11]

Other neuromechanical adaptations are the evoked spinal reflexes assessed by Ia-afferent-mediated H-reflex and the net excitation of the neuron pool determined by the first volitional wave (V-wave).[15] Higher V-wave but not H-reflex values of the soleus muscle have been observed in the affected legs of athletes with chronic middle-portion AT.[16] This greater V-wave may result from an enhanced neural drive in the descending corticospinal pathways, elevated net excitability of both large and small motor neurons, and/or alterations in the presynaptic inhibition during voluntary activation of the soleus muscle.[17] However, a recent study using a similar approach has shown higher H-reflex and V-wave values in individuals

with mid-portion AT compared with controls,[11] showing that there is some discrepancy across studies.

Once symptoms develop, ensuing movement dysfunction may contribute to the chronicity of symptoms.[18] Pain in the Achilles tendon causes widespread motor inhibition in the affected region, evidenced by the lower electrical activity of the agonist, synergist and antagonist muscles.[19] Individuals with tendinopathy also tend to use movement patterns that place excessive or abnormal load on their tendons; the faulty movement may represent either a root cause or a reason for chronicity or slow resolution.[18] This may be attributed to a protective mechanism that prevents further injury or even tendon rupture.[20]

Research on the treatment of AT is somewhat scarce despite the prevalence.[21] Over the past decade, loading-based treatment in the form of eccentric training (exercises where tendon lengthening during active contractions is emphasised) has become the main non-surgical choice of treatment for AT,[22] although there is no convincing evidence showing that this form of exercise is the most effective for AT. A recent systematic review concluded that there is little clinical or mechanistic evidence supporting the use of isolated eccentric exercises alone.[23] Well-conducted studies comparing different loading programmes are largely lacking.[23] The purpose of exercise is to provide mechanical loading to the tendon in order to promote remodelling, decrease pain and improve calf-muscle endurance and strength.[6 24] It seems that loading itself yields positive clinical, structural and biochemical effects with respect to tendinopathy.[6 25–28] However, the successful management of AT remains challenging, possibly due to a lack of knowledge about the effect of loading parameters that is, load progression, load magnitude, frequency (sets and repetitions) and restitution between treatment sessions.[21]

There are several systematic reviews about the effects of exercise in individuals with AT. However, most are focused on pain and function and usually only use self-reported outcome measures as a main outcome. Self-reported outcomes have high variability among the population, thus, the conclusion of these studies may be partially biased. Although two recent systematic reviews explored the effects of exercise on the morphological properties of the Achilles tendon in individuals with mid-portion AT,[29 30] to our knowledge, there are currently no systematic reviews examining the neuromechanical changes that occur in individuals with AT or the effects of exercise on these properties. Therefore, the aim of this systematic review is to synthesise the current literature regarding (1) triceps surae–Achilles tendon complex neuromechanics in individuals with AT and (2) the effect of exercise-induced mechanical tendon loading on neuromechanical changes induced by AT.

## METHODS

This systematic review protocol has been developed following the Cochrane Handbook for Systematic

 Contreras-Hernandez I, *et al. BMJ Open* 2022;**12**:e050186. doi:10.1136/bmjopen-2021-050186

Reviews of Interventions and the Preferred Reporting Items for Systematic Review and Meta-analysis Protocols (PRISMA-P) 2015 checklist (online supplemental file 1).[31–33] This protocol has been registered with the International Prospective Register of Systematic Reviews (PROSPERO) on 26 January 2021 (registration number: CRD42021231933). The Cochrane Handbook mainly focuses on the synthesis of evidence from intervention studies which is related only to the second objective of this systematic review; unfortunately, the first objective is commonly addressed by observational studies. Therefore, we followed these guidelines in addition to the COSMOS-E guidance for the following reasons: First, there are many similarities in the general structure and procedures used in both types of reviews. Second, the information in the Cochrane Handbook is described in more detail and includes important information related to observational studies. Third, a similar approach has been made in other systematic review protocols.[34 35] Fourth, widely accepted standards of systematic reviews meta-analysis of observational studies are lacking.[36] We know that some methodological differences exist, but we will pay particular attention to certain steps of the conducted process (eg, choice of statistical methods, sources of heterogeneity, etc).[36]

### Eligibility criteria

The PICOS framework (Population, Intervention, Comparison, Outcomes, and Study Design) has been used to define the eligibility criteria for the inclusion and exclusion of studies in this systematic review.[32 37] However, due to the characteristics of this research, we will use 'Indicator' in the same category of 'Intervention' as it has been used previously.[34 35]

### Population

The population of interest is adults (aged 18–65 years) with mid-portion or insertional AT and pain-free adults as a control group. However, we will also include studies that have used the asymptomatic lower limb as a control. In order to avoid excluding relevant articles identified during the scoping search, participants with bilateral AT will also be included. There will be no restrictions in terms of gender or ethnicity. Studies that include individuals with AT who have been diagnosed with an underlying medical pathology or disorder (eg, systemic inflammatory conditions, cardiovascular diseases, neurological disorders) and/or history of Achilles tendon surgery will be excluded.

### Intervention/indicator

In order to address the first aim of this systematic review, eligible studies will be those which include the use of any electrophysiological technique (eg, surface electromyography, intramuscular electromyography, high-density surface electromyography, transcranial magnetic stimulation, cervicomedullary magnetic stimulation or peripheral nerve stimulation) to determine neuromuscular,

spinal, subcortical or supraspinal changes in people with AT. Additionally, eligible studies will be those which include the use of ultrasonographic or MRI to measure the morphological or mechanical properties of the Achilles tendon in people with AT. Similarly, eligible studies to address the second aim of this systematic review will be those which include the use of any electrophysiological technique to determine neuromuscular, spinal, subcortical or supraspinal changes produced by any exercise-induced mechanical tendon loading protocols (eg, eccentric, isometric or concentric contractions, plyometric exercises, stretching or rehabilitation protocols) in people with AT. Moreover, eligible studies will be those which include the use of ultrasonographic or MRI to measure changes in the morphological or mechanical properties of the Achilles tendon produced by any exercise-induced mechanical loading in individuals with AT.

### Comparison

Studies must include a comparison of the neuromuscular properties of the triceps surae muscle or morpho-mechanical properties of the Achilles tendon between individuals with AT and controls or between symptomatic and asymptomatic lower limbs. Likewise, studies assessing the effects of exercise-induced mechanical tendon loading should include a comparison of the neuromuscular features of the triceps surae muscle or morpho-mechanical properties of the Achilles tendon in individuals with AT and controls or between symptomatic and asymptomatic lower limbs.

### Outcomes

Primary outcomes will include neuromuscular properties of the triceps surae muscle and morpho-mechanical features of the Achilles tendon. We will include studies assessing the amplitude and timing of EMG activity of the gastrocnemius-soleus (millivolts and milliseconds, respectively), tibialis anterior and gastrocnemius-soleus coactivation (co-contraction ratio, %), gastrocnemius-soleus motor-evoked potentials (MEPs, microvolts) obtained from transcranial and cervicomedullary magnetic stimulation (TMS and CMEPs), H-reflex (peak-to-peak amplitude, microvolts), F-wave or V-wave obtained by peripheral nerve stimulation (tibial nerve) and motor unit data (motor unit discharge rate, Hz) obtained from intramuscular and/or high-density surface EMG recordings (millivolts). The present systematic review will also include morpho-mechanical properties such as length (cm), thickness (mm), CSA ($mm^2$), volume ($cm^3$), stiffness (N/mm or kPa), modulus (kPa), creep, elasticity (kPa), strain (%) and stress (kPa) of the Achilles tendon. Only studies that measure any of these neuromechanical properties quantitatively will be included. Secondary outcomes will include duration of symptoms (months), the severity of symptoms (Victorian Institute of Sports Assessment-Achilles questionnaire (VISA-A) or Visual Analogue Scale (VAS)), type of tendinopathy (mid-portion, insertional

or both), diagnostic confirmation (clinical assessment and/or ultrasound evaluation) and physical activity level (International Physical Activity Questionnaire (IPAQ), hours of physical activity per week, etc).

## Study design

Based on scoping searches, randomised controlled trials and non-randomised controlled trials (ie, cohort, cross-sectional and cohort studies) will be considered to address both objectives of this systematic review adequately. Non-original literature (eg, systematic and narrative reviews) or other types of studies will be excluded and reported in the PRISMA flow diagram.

## Information sources

The following electronic databases will be used from inception to February 2021: Pubmed, MEDLINE (Ovid Interface), EMBASE (Ovid Interface), CINAHL Plus (EBSCO Interface), Web of Science (WOS; Clarivate Analytics) and SPORTDiscus (EBSCO Interface). Specific research strategies have been designed considering medical subject heading (MESH) terms to improve search results. Moreover, hand searching of key journals will be conducted, including *Journal of Applied Physiology, Journal of Orthopaedic & Sports Physical Therapy, Journal of Electrophysiology and Kinesiology, Journal of Biomechanics, Clinical Biomechanics, British Journal of Sports Medicine, Medicine and Science in Sports and Exercise, Journal of Science and Medicine in Sports,* and *Isokinetic and Exercise Science.* The eligibility of the manuscripts found in hand searching will be defined using the PICOS framework. Additionally, relevant authors in the field will be contacted to identify unpublished articles in preparation. To minimise risk of bias publication, grey literature will be also included, and searches will be conducted using the British national bibliography for report literature (BNBRL), ProQuest Dissertations & These Global, OpenGrey database and EThOs. Reference lists of included studies and relevant systematic reviews will be checked for any further studies, accordingly with the MECIR standards.[38]

## Search strategy

Two independent reviewers (IC-H and AS) will complete the search and identify potential studies to be included in this systematic review. There will be no restrictions in terms of date, design or language, to ensure inclusion of all relevant articles.

Due to the inability to obtain maximal retrieval of articles during the scoping search and to adequately address both objectives of this systematic review, this search will be conducted in a two-step process:

1. Initial search to identify studies with neuromuscular properties of the triceps surae or morpho-mechanical features of the Achilles tendon in individuals with AT.
2. Secondary search identifying studies assessing the effects of exercise-induced tendon mechanical loading on neuromuscular properties of the triceps surae or

the morpho-mechanical characteristics of the Achilles tendon in individuals with AT.

A search strategy example for MEDLINE (Ovid Interface) database of each step is reported in online supplemental file 2 and includes MESH terms, keywords and search strings to ensure maximal retrieval.[39] The specific search terms will be modified to reflect differences in keywords and syntax between databases but search strategy consistency will be guaranteed.

## Data management

Literature search results, including citation and abstract of potentially eligible studies, will be imported into EndNote V.X9 (Clarivate Analytics PCL) reference manage software by one reviewer (IC-H), allowing the identification and removal of any duplicates before the screening process. Abstracts and full texts of potentially eligible studies will be saved in an individual folder for each reviewer (IC-H, AS) and eligible studies will be retrieved and stored in EndNote V.X9. To effectively accomplish the screening process, forms that have been developed to reflect the inclusion and exclusion criteria will be used.

## Study selection

Before the screening process, screening forms will be tested by two reviewers (IC-H, AS) in a small number of articles to ensure their effectiveness. The screening process will then start with the assessment of titles and abstracts of identified studies by two reviewers (IC-H, AS), and they will subcategorise them into definitely eligible, definitely ineligible or doubtful.[40] In the event of disagreement, reviewers will first attempt to resolve through discussion; however, if consensus cannot be reached, a third reviewer (EM-V) will mediate the process. Then, the reviewers will perform full-text screening of potentially eligible studies independently. Similarly, if no consensus is possible, a third reviewer will support the process. The agreement between the reviewers during both screening stages will be determined using the kappa statistic, and the PRISMA flow diagram will be used to summarise the study selection process.[32]

## Data collection process

The data collection process will begin developing a standardised form based on the Cochrane data extraction template, objectives of the systematic review and inclusion criteria as a guide. A standardised form will be piloted a priori on a subgroup of studies. IC-H will extract data, and AS will check the accuracy of this process. Any discrepancies will be discussed between the two reviewers; however, the third reviewer (EM-V) will determine which data are relevant if no agreement is achieved. Authors of the primary studies will be contacted if any critical information that needs to be extracted is missing. If multiple publications of the same study exist, they will be collated, the primary authors contacted for further clarification and the duplicates removed. Likewise, if potentially eligible studies appear

**Table 1** Characteristics of included studies

| Information about data | Data extracted |
|---|---|
| Date of data extraction | |
| General study information | Title<br>Authors<br>Year of publication |
| Study methodology | Study design<br>Sample size<br>Individuals' characteristics (age, gender, weight, height, physical activity level, etc)<br>Diagnostic confirmation (clinical evaluation, ultrasound/MRI assessment, use of questionnaires, etc)<br>Achilles tendinopathy group characteristics (location, side, pain intensity, duration of symptoms, etc)<br>Type of instrument used to measure the neuromuscular properties<br>Type of instrument used to determine the morpho-mechanical properties (ultrasonography or MRI)<br>Type of exercise-induced mechanical tendon loading protocol applied |
| Outcome | The neuromuscular properties include the amplitude and timing of EMG activity of the gastrocnemius-soleus, tibialis anterior and gastrocnemius-soleus coactivation, gastrocnemius-soleus MEPs, H-reflex, F-wave and motor unit data.<br>The morpho-mechanical properties of the Achilles tendon include length, thickness, cross-sectional area, volume, stiffness, modulus, creep, elasticity, strain and stress.<br>The comparison could be within groups (eg, affected vs non-affected side) or between groups (eg, Achilles tendinopathy group vs control group). |
| Funding, declaration of conflict of interest | Funding information<br>Conflict of interest of authors |

MEPs, motor-evoked potentials.

to use the same data during the data collection process, the primary authors will be contacted, and a specific report will be selected. The decision of the selected report will be justified.

## Data items

Data items to be extracted include general study information, participants' characteristics, measurement methods and outcome measures. These items are presented in table 1. We will use the same extraction sheet for each step of the systematic review; the only difference will be the item regarding 'Type of exercise-induced mechanical tendon loading protocol applied'. If any eligible studies include more than two groups, data will be extracted only from the control group and the one that meets the eligibility criteria.

## Risk of bias

The risk of bias will be determined independently by two reviewers (IC-H and AS) using the RoB 2 and ROBINS-I tools according with Cochrane recommendations.[31] The RoB 2 is a tool to determine the risk of bias in randomised trials and includes the following domains: bias arising from the randomisation process, bias due to deviations from intended interventions, bias due to missing outcome data, bias in measurement of the outcome, bias in selection of the reported result and overall bias.[41] The domains were selected to address all important mechanisms by which bias can be introduced into the results of a trial, based on a combination of empirical evidence and theoretical considerations.[41] Each domain is required, and no additional domains should be added. For each domain, the tool comprises a series of 'signalling questions', a judgement about risk of bias for the domain, free text boxes to justify responses to the signalling questions and risk-of-bias judgement, and an option to predict (and explain) the likely direction of bias.[31] Signalling questions aim to elicit information relevant to an assessment of risk of bias.[41] The questions seek to be reasonably factual in nature.[41] The response options are 'yes,' 'probably yes,' 'probably no,' 'no' and 'no information'.[41] Based on these responses, the options for a domain-level risk-of-bias judgement are 'Low', 'Some concerns' and 'High' risk of bias.[41]

The ROBINS-I will be used to determine the risk of bias in non-randomised studies of interventions and include the following domains: bias due to confounding, bias in selection of participants into the study, bias in classification of interventions, bias due to deviations from intended interventions, bias due to missing data, bias in measurement of outcomes and bias in the selection of the reported result.[42] Each domain is mandatory, and no additional domain should be added. The tool comprises, for each domain, a series of 'signalling questions', a judgement about risk of bias for the domain, free text boxes to justify responses to the signalling questions and risk-of-bias judgements and an option to predict (and explain) the likely direction of bias.[31] The signalling questions aim to elicit information relevant to the risk-of-bias judgement for the domain, and work in the same way as for RoB 2.[31] The response options are 'yes,' 'probably yes,' 'probably no,' 'no' and 'no information'.[42] Based on these responses, the options for a domain-level risk-of-bias judgement are 'Low', 'Moderate', 'Serious' or 'Critical' risk of bias, with an additional option of 'no information'.[42]

Disagreements between the reviewers regarding the risk of bias in a study will be resolved by discussion, with the involvement of a third review author (EM-V) if necessary.

## Data synthesis

A meta-analysis will be considered if outcomes and methodology of the selected studies are homogeneous. If possible, the two reviewers (IC-H, AS) will independently

group studies that are more homogeneous using the following characteristics:

► Parameter used to determine neuromuscular properties of the triceps surae.
► Parameter used to measure morpho-mechanical properties of the Achilles tendon.
► Type of exercised-induced mechanical tendon loading protocol applied.

Disagreement between the reviewers will be resolved by discussion, but if no agreement is possible, a third reviewer (EM-V) will mediate the process.

Whether clinical or methodological homogeneity across studies investigating the same outcome domain is sufficient, statistical heterogeneity will be performed. The amount of inconsistency among studies will be assessed using the $I^2$ statistic.[43] As in previous reviews, the grouping of studies will be eligible for meta-analysis if an $I^2$ value of <50% (low heterogeneity) is determined.[44] Then, a random-effects meta-analysis will be performed for each subgroup following the recommendations of Deeks *et al.*[45] Extracted data will be converted into a common rubric, which most likely be ORs with 95% CIs since we are dealing mostly with binary data.[46] If the data are not sufficiently homogeneous, the results of the considered outcomes will be described using the vote-counting procedure (direction of difference or no difference) and a narrative synthesis will be developed.[47] The narrative synthesis will be conducted following the recommendations of Popay and Snowden.[46]

### Confidence in cumulative evidence

Data pooled quality (certainty) will be assessed using the Grading of Recommendations Assessment, Development, and Evaluation (GRADE) approach.[48 49] This process include five steps described by Goldet and Howick,[50] and the final quality of evidence will be presented as 'High', 'Moderate', 'Low' or 'Very Low'. The certainty of evidence for each outcome across studies can be decreased by risk of bias, inconsistency, indirectness, imprecision and publication bias.[51] Conversely, certainty of evidence can be increased by large effect size, dose–response gradient and plausible confounding biases that underestimate the effect size.[51] Finally, recommendations for the interpretation of the evidence quality will be given, following the criteria of Guyatt *et al.*[52]

### Patient and public involvement

The topic of this systematic review protocol was not discussed at our established patient and public involvement meetings (PPI), due to COVID-19 pandemic. Patients will not be involved in the analysis and data collection of this project, but our results will be presented at PPI meetings at the University of Birmingham in the future.

### Ethics and dissemination of results

Ethical approval is not required for this review, as it will only involve the collation of previously published data.

Our findings will be disseminated through publication in a peer-reviewed journal and presented at national and/or international conferences.

**Contributors** IC-H and EM-V are responsible for the conception of the research question and development of the protocol. IC-H wrote the first draft of the protocol with guidance from EM-V and DF. IC-H and AS will be the first and second reviewers, respectively. EM-V will be the third reviewer. All drafts were revised and reviewed by all the authors before the final approval of the last version of the manuscript.

**Funding** The authors have not declared a specific grant for this research from any funding agency in the public, commercial or not-for-profit sectors.

**Competing interests** None declared.

**Patient consent for publication** Not applicable.

**Provenance and peer review** Not commissioned; externally peer reviewed.

**ORCID iDs**
Deborah Falla http://orcid.org/0000-0003-1689-6190
Alessandro Schneebeli http://orcid.org/0000-0002-8411-2012
Eduardo Martinez-Valdes http://orcid.org/0000-0002-5790-7514

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
