## [Reviewer comments · BMJ Open]

ARTICLE DETAILS

TITLE (PROVISIONAL)	Neuromechanical changes in Achilles tendinopathy and the effects of exercise-induced mechanical tendon loading. A protocol for a systematic review.
AUTHORS	Contreras-Hernandez, Ignacio; Falla, Deborah; Schneebeli, Alessandro; Martinez-Valdes, Eduardo

VERSION 1 – REVIEW

REVIEWER	Papalia, R Campus Biomedico University of Rome
REVIEW RETURNED	21-Apr-2021

GENERAL COMMENTS	Dear Authors, the protocol you submitted is interesting in its scope, as it deals with a peculiar aspect of Achilles pathology. A few comments: The introduction could be improved as it fails to describe the rationale that lead you to perform such a specific review. The neuromechanics of AT is not well understood and known to the large public, thus a thorough summery of the available evidence on the topic, maybe showing the contrasting available results, could help in understand the aim of your review. The search strategy should be described in the most reproducible way. I suggest to report in the supplementary file 2 the search strings, rather than isolated MeSH and groups of keywords. Last, did you planned any methanalysis on potential quantitative outcomes reported in the eligible studies?
---

REVIEWER	Usuelli , Federico Galeazzi Orthopedic Institute for Research and Care
REVIEW RETURNED	30-Apr-2021

GENERAL COMMENTS	Dear Authors, Thank you for the manuscript you submitted and for the opportunity to make revisions. I appreciated the correct methodology, the valid premises, the clarity of the description, the description of clinical and scientific background. In my opinion, is particularly valuable the deductive method you applied to describe from the general to the particular, with a varied examination that deals with the definition of tendinopathies to the specific case, passing through the pathophysiology of AT.
--

	However, I list my considerations on the manuscript below.  - lines 68-73: the sentence is too long, I recommend checking the punctuation and a better highlighting points 1) and 2), which are the subject of your protocol - line 70: you mention mid-portion; instead, line 107: mid-portion and insertional lesions are included. I recommend to clarify the distinction between the two aspects in the introduction, furthermore to clarify better in the scope of the protocol (lines 73-75) and in the inclusion criteria which pathology you want to specifically study (also evaluating whether to treat both and insert a sub-section in the study protocol) - line 155: "31 February 2021". I recommend replacing it with "February 2021" or "February 28 2021". In conclusion, I believe that the protocol presented is an ambitious research project that could bring benefits in terms of knowledges to a large number of colleagues, especially young orthopedists who have recently approached the study of tendinopathies.
--	--

REVIEWER	Belloti, João Federal University of São Paulo (UNIFESP/EPM), Orthopedics and Traumatology - Division of Hand Surgery and Upper Limb
REVIEW RETURNED	07-May-2021

GENERAL COMMENTS	Although the authors described using the Cochrane methodology of systematic reviews of intervention in the elaboration of this protocol, there are several serious flaws in the proposed study.  1- There is no clear description of intervention and control to be adopted in the inclusion of studies. 2- The authors describe the inclusion of all types of experimental and observational study designs, an inadequate condition for systematic intervention reviews: "All types of experimental and observational studies using quantitative methods will be eligible for inclusion (including randomized clinical trials, non-randomized clinical trials, cohort, case-control, cross-sectional studies, case series, and case reports) (PROSPERO CRD42021231933)". 3- There is no description of the primary and secondary outcomes of the study, which prevents their reproducibility and interpretation of results.
--

VERSION 1 – AUTHOR RESPONSE

Reviewer: 1
Dr. R Papalia, Campus Biomedico University of Rome
Comments to the Author:
Dear Authors,

the protocol you submitted is interesting in its scope, as it deals with a peculiar aspect of Achilles pathology.

A few comments:

The introduction could be improved as it fails to describe the rationale that lead you to perform such a specific review. The neuromechanics of AT is not well understood and known to the large public, thus

a thorough summary of the available evidence on the topic, maybe showing the contrasting available results, could help in understand the aim of your review.

Following your recommendations, we have improved our introduction including additional information about the basic of neuromechanics and recent evidence about the neuromechanical properties in individuals with Achilles tendinopathy.

We added the following paragraphs to the introduction:

“Human movement emerges from the interplay among descending output from the central nervous system (CNS), sensory input from the body and environment, muscle dynamics and the whole body dynamics (Nishikawa et al., 2007). Thus, neuromechanics is the study of the coupling between neural information processing and mechanical behavior (Nishikawa et al., 2007). Fundamentally, the CNS plans, initiates and sends motor commands to muscle (Wolpert, 2013) then, the muscle executes the motor command producing force to pull the tendon, and finally, the tendon transmits and modulates muscle force controlling the movement (Chang and Kulig, 2015). Additionally, peripheral components of this hierarchical system (i.e., muscle spindles and Golgi tendon organs) send feedback signals to assist the CNS in motor command planning (Scott, 2004).”

“Other neuromechanical adaptations are the evoked spinal reflexes assessed by Ia-afferent-mediated H-reflex and the net excitation of the neuron pool determined by the first volitional wave (V-wave) (Knikou, 2008). Higher V-waves but not H-reflex values of the soleus muscles have been observed in the affected legs of athletes with chronic middle-portion Achilles tendinopathy (Wang et al., 2011). This greater V-wave may result from an enhanced neural drive in the descending corticospinal pathways, elevated net excitability of both large and small motor neurons, and/or alterations in the presynaptic inhibition during voluntary activation of the soleus muscle (Aagaard et al., 2002). However, a recent study using a similar approach has shown higher H-reflex and V-wave values in individuals with mid-portion AT compared with controls (Chang and Kulig, 2015), showing that there is some discrepancy across studies”

The search strategy should be described in the most reproducible way. I suggest to report in the supplementary file 2 the search strings, rather than isolated MeSH and groups of keywords.

Due to the decision to perform this systematic review as a two-steps process, we have included the MeSH terms, groups of keywords and full search string of each search strategy as a supplementary file (supplementary file 2).

Last, did you planned any meta-analysis on potential quantitative outcomes reported in the eligible studies?

As we stated in the data synthesis section, a meta-analysis will be considered if the outcomes and methodology of the selected studies are homogenous. Based on our scoping searches, we believe that we will be able to perform a meta-analysis to address the first aim of this systematic review.

Reviewer: 2

Dr. Federico Usuelli , Galeazzi Orthopedic Institute for Research and Care

Comments to the Author:

Dear Authors,

Thank you for the manuscript you submitted and for the opportunity to make revisions.

I appreciated the correct methodology, the valid premises, the clarity of the description, the description of clinical and scientific background.

In my opinion, is particularly valuable the deductive method you applied to describe from the general to the particular, with a varied examination that deals with the definition of tendinopathies to the specific case, passing through the pathophysiology of AT.

However, I list my considerations on the manuscript below.

- lines 68-73: the sentence is too long, I recommend checking the punctuation and a better highlighting points 1) and 2), which are the subject of your protocol

Considering your suggestions, we have modified the introduction to more clearly state the aims of this systematic review.

We have modified the following paragraph of the introduction:

“There are several systematic reviews about the effects of exercise in individuals with AT. However, most are focused on pain and function and usually only use self-reported outcome measures as a main outcome. Self-reported outcomes have high variability among the population, thus, the conclusion of these studies may be partially biased. Although two recent systematic reviews explored the effects of exercise on the morphological properties of the Achilles tendon in individuals with mid-portion AT (Merza et al., 2021; Färnqvist et al., 2020) to our knowledge, there are currently no systematic reviews examining the neuromechanical changes that occur in individuals with AT or the effects of exercise on these properties. Therefore, the aim of this systematic review is to synthesise the current literature regarding 1) Triceps surae-Achilles tendon complex neuromechanics in individuals with AT and 2) The effect of exercise-induced mechanical tendon loading on neuromechanical changes induced by AT.”

- line 70: you mention mid-portion; instead, line 107: mid-portion and insertional lesions are included. I recommend to clarify the distinction between the two aspects in the introduction, furthermore to clarify better in the scope of the protocol (lines 73-75) and in the inclusion criteria which pathology you want to specifically study (also evaluating whether to treat both and insert a sub-section in the study protocol)

Regarding your second comment, we mentioned two other systematic reviews that have explored the effects of exercise on the Achilles tendon morphological properties in individuals with mid-portion Achilles tendinopathy. However, our systematic review will include studies with mid-portion and insertional Achilles tendinopathy because, after our scoping searches, we noticed that several studies include individuals with Achilles tendinopathy in general, without distinguishing between mid-portion or insertional. Additionally, we have decided to add a brief description of the differences between mid-portion and insertional Achilles tendinopathy in the introduction.

The information that we added to the introduction can be found below:

“Achilles tendon injuries can be separated into non-insertional tendinopathy (55%-65% of the injuries) insertional tendinopathy (20%-25%), and proximal musculotendinous junction injuries (9-25%), according to the location of pain (Kvist, 1991). Nevertheless, individuals may present symptoms at the insertion and mid-portion concurrently, and approximately 30% have bilateral pain (Silbernagel et al., 2007)”

- line 155: "31 February 2021". I recommend replacing it with "February 2021" or "February 28 2021". According with your recommendation, we have replaced the date with “February 2021”

In conclusion, I believe that the protocol presented is an ambitious research project that could bring benefits in terms of knowledges to a large number of colleagues, especially young orthopedists who have recently approached the study of tendinopathies.

Reviewer: 3

Dr. João Belloti, Federal University of São Paulo (UNIFESP/EPM), Universidade Federal de São Paulo Escola Paulista de Medicina

Comments to the Author:

Although the authors described using the Cochrane methodology of systematic reviews of intervention in the elaboration of this protocol, there are several serious flaws in the proposed study.

1- There is no clear description of intervention and control to be adopted in the inclusion of studies.

Regarding your first comment, we know that the Cochrane methodology of systematic reviews mainly focuses on the synthesis of evidence from intervention studies which is related only to the second objective of this systematic review. However, we followed this guideline in addition to the COSMOS-E

guidance for the following reasons: First, there are many similarities in the general structure and procedures of both types of guidelines. Second, the information in the Cochrane Handbook is described in more detail and includes important information related to observational studies. Third, a similar approach has been made in other systematic review protocols (Arvanitidis et al., 2021; Devecchi et al., 2019). Fourth, widely accepted standards of systematic reviews meta-analysis of observational studies are lacking. We are aware of the methodological differences, but we will pay particular attention to specific steps of the systematic review process (e.g., choice of statistical methods, sources of heterogeneity, etc.). Accordingly, to clearly state the Intervention/Indicator of each step of the systematic review, we have modified this section.

Changes to this section can be found below:

Intervention/Indicator

“In order to address the first aim of this systematic review, eligible studies will be those which include the use of any electrophysiological technique (e.g., surface electromyography, intramuscular electromyography, high-density surface electromyography, transcranial magnetic stimulation, cervicomedullary magnetic stimulation, or peripheral nerve stimulation) to determine neuromuscular, spinal, subcortical, or supraspinal changes in people with AT. Additionally, eligible studies will be those which include the use of ultrasonographic or Magnetic Resonance Imaging (MRI) to measure the morphological or mechanical properties of the Achilles tendon in people with AT. Similarly, eligible studies to address the second aim of this systematic review will be those which include the use of any electrophysiological technique to determine neuromuscular, spinal, subcortical, or supraspinal changes produced by any exercise-induced mechanical tendon loading protocols (e.g., eccentric, isometric, or concentric contractions, plyometric exercises, stretching or rehabilitation protocols) in people with AT. Moreover, eligible studies will be those which include the use of ultrasonographic or MRI to measure changes in the morphological or mechanical properties of the Achilles tendon produced by any exercise-induced mechanical loading in individuals with AT.”

2- The authors describe the inclusion of all types of experimental and observational study designs, an inadequate condition for systematic intervention reviews: "All types of experimental and observational studies using quantitative methods will be eligible for inclusion (including randomized clinical trials, non-randomized clinical trials, cohort, case-control, cross-sectional studies, case series, and case reports) (PROSPERO CRD42021231933) ".

We apologize for this mistake, and we thank the reviewer for reminding us to update the register in PROSPERO. The register has now been updated. In the study design section of the registry, it is now stated:

"Based on scoping searches, randomised controlled trials and non-randomised controlled trials (i.e., cohort, cross-sectional, and cohort studies) will be considered to address both objectives of this systematic review adequately. Non-original literature (e.g., systematic and narrative reviews) or other types of studies will be excluded and reported in the PRISMA flow diagram."

3- There is no description of the primary and secondary outcomes of the study, which prevents their reproducibility and interpretation of results.

Following your recommendations, we have established primary and secondary outcomes with their respective unit of measurement to allow reproducibility and interpretation of results.

Primary and secondary outcomes can be found below:

Outcomes

“Primary outcomes will include neuromuscular properties of the triceps surae muscle and morpho-mechanical features of the Achilles tendon. We will include studies assessing the amplitude and timing of EMG activity of the gastrocnemius-soleus (millivolts and milliseconds, respectively), tibialis anterior and gastrocnemius-soleus co-activation (co-contraction ratio, %), gastrocnemius-soleus

motor evoked potentials (MEPs, microvolts) obtained from transcranial and cervicomedullary magnetic stimulation (TMS and CMEPs), H-reflex (peak-to-peak amplitude, microvolts), F-wave or V-wave obtained by peripheral nerve stimulation (tibial nerve) and motor unit data (motor unit discharge rate, Hz) obtained from intramuscular and/or high-density surface EMG recordings (millivolts). The present systematic review will also include morpho-mechanical properties such as length (cm), thickness (mm), cross-sectional area (mm²), volume (cm³), stiffness (N/mm or kPa), modulus (kPa), creep, elasticity (kPa), strain (percentage), and stress (kPa) of the Achilles tendon. Only studies that measure any of these neuromechanical properties quantitatively will be included. Secondary outcomes will include duration of symptoms (months), severity of symptoms (Victorian Institute of Sports Assessment-Achilles questionnaire (VISA-A) or Visual Analog Scale (VAS)), type of tendinopathy (mid-portion, insertional or both), diagnostic confirmation (clinical assessment and/or ultrasound evaluation), and physical activity level (International Physical Activity Questionnaire (IPAQ), hours of physical activity per week, etc.)."